# Hierarchical Multi-Scale Modeling of Absolute Binding Affinity in Protein Complexes

**Dongyun Kim**[1,2,*,†]**& Soorin Yim**[1,*]**& Sungjoon Park**[1] **& Kiwoong Yoo**[1] **& Kyungwook Lee**[1]
**Doyeong Hwang**[1] **& Soonyoung Lee**[1] **& Jongseong Jang**[1] **& Kiyoung Kim**[1]
[1]LG AI Research [2]School of Biological Sciences, Seoul National University
dy.kim@snu.ac.kr,{soorin.yim, elgee.kim}@lgresearch.ai

## Abstract

Predicting the absolute binding affinity ($\Delta G$) of a protein-protein complex from structure alone remains challenging: physics-based simulations are costly, and experimental affinity labels are noisy and heterogeneous. We propose a hierarchical architecture that represents each complex at multi-scale resolutions—atoms, residues, chains, and the whole complex. We use distance-weighted message passing so that closer atom/residue pairs contribute more strongly, and then pool information across levels to produce a single binding affinity score. The model further incorporates transferable physicochemical priors via pretrained representations. On PPB-Affinity dataset, our method improves rank correlation over a strong baseline (Spearman's rank correlation coefficient 0.659 vs. 0.646). Ablations show that distance-weighted message passing, multi-scale modeling, and physically grounded representations each contribute to model performance.

## 1 Introduction

Protein-protein interactions (PPIs) are central to cellular function, and their binding affinity is governed by the binding free energy, $\Delta G$. In *de novo* protein design, estimating $\Delta G$ is crucial because absolute thermodynamic stability—rather than relative improvements of mutants over a known wild-type reference—ultimately determines whether a designed protein can fold and bind into an intended complex. Physics-based approaches such as molecular dynamics simulations or Molecular Mechanics/Poisson–Boltzmann Surface Area (MM/PBSA) (Kollman et al., 2000) are principled, but often computationally expensive and sensitive to sampling. Empirical scoring functions such as FoldX (Delgado et al., 2019) are efficient yet their reliance on static structures and simplified potentials limits their ability to capture the energetic complexity of *de novo* interfaces.

Deep learning offers an opportunity to predict binding affinities directly from protein structures without explicit simulation. Recently, advances in structure prediction and sequence design models such as AlphaFold (Abramson et al., 2024) and ProteinMPNN (Dauparas et al., 2022) have shifted emphasis away from explicit energetic modeling. In practice, model confidence scores are often used as a naive proxy for binding affinity, but their accuracy remains limited. Alongside, because absolute binding labels are scarce, noisy, and assay-dependent, much prior work has shifted toward predicting mutational changes in binding affinity, $\Delta\Delta G = \Delta G_{\text{mutant}} - \Delta G_{\text{wild-type}}$ (Bushuiev et al., 2024; Cai et al., 2024), which is easier to learn and partially cancels systematic measurement bias across experimental settings. Yet, $\Delta\Delta G$ is inherently reference-dependent, limiting its utility in *de novo* design where no wild-type reference complex exists and absolute folding and binding stabilities must be assessed for plausible design filtering. Although recent large-scale datasets have renewed interest in learning absolute $\Delta G$ (Liu et al., 2024; Tsuboyama et al., 2023), many existing models still rely on monomeric representations or are restricted to single-assay training to avoid cross-source noise (Lee et al., 2026).

To address these limitations, we propose a hierarchical multi-scale graph framework that enables robust learning of absolute binding affinity from multimeric complex structures. Our contributions are as follows:

---

[*]Equal contribution.
[†]Work done during an internship at LG AI Research.

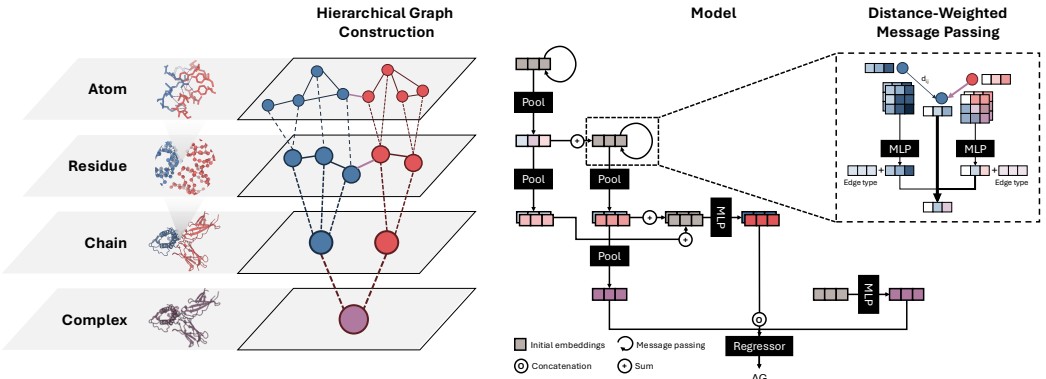

Figure 1: Hierarchical multi-scale architecture for predicting absolute binding affinity. A complex is represented as a four-level graph (atoms, residues, chains, complex); atom/residue message passing uses distance-weighted edges and is pooled downward to chain embeddings. Residue, chain, and complex embeddings are concatenated to produce the final predictions.

- **Robustness:** To mitigate label inconsistencies from experimental noise and assay-specific biases, we adopt data relabeling to construct self-consistent $\Delta G$ targets. We further train the model with regression and auxiliary pairwise ranking objectives to improve robustness.

- **Multi-scale:** We enhance structural resolution by representing each complex through four coupled hierarchical levels (atoms, residues, chains, and the global complex). This multi-resolution approach, integrated via distance-weighted message passing, enables the model to discern intricate interfacial energetics, unlike $\Delta\Delta G$ prediction, which often relies on localized mutational changes between simplified representations.

- **Ablation study:** Beyond achieving state-of-the-art performance, we provide a deeper analysis of the contribution of each structural resolution. Our findings demonstrate that multi-scale integration is essential for capturing the full spectrum of binding thermodynamics, from local chemical contacts to macro-molecular entropic effects, offering new insights into the structural basis of binding affinity.

## 2 METHODS

We propose HierBind, a Hierarchical multi-scale Message Passing Neural Network for predicting absolute binding affinity from a protein complex structure. As shown in Figure 1, we build a multi-scale interaction graph, perform distance-weighted message passing at the atom and residue levels, and then hierarchically pool representations. Finally, we concatenate pooled residue-, chain-, and complex-level representations and predict an affinity score with a regressor.

### 2.1 HIERARCHICAL GRAPH FEATURES

We represent each complex as a coupled hierarchy of four node types: atoms, residues, chains (ligand and receptor), and a global complex node. We build the atom- and residue-level graphs in 3D space using $k$-nearest neighbors ($k = 16$) based on Euclidean distance. For residue-residue distances, we use $C\alpha$ coordinates.

We initialize each node with a learned 32-dimensional type embedding (atom element for atom nodes and amino-acid identity for residue nodes). Chain and complex nodes are initialized as zeros. When available, we augment these initial embeddings by adding pretrained ATOMICA representations at the corresponding level to inject transferable physicochemical priors (Fang et al., 2025); for chain nodes, we use ATOMICA global block representations.

Each edge is annotated with an edge-type embedding (intra-chain vs. inter-chain) and a distance-dependent weight. We discretize distances into 8 bins and apply a monotone decreasing weighting

scheme so that closer neighbors contribute more strongly. Inter-chain edges are additionally scaled by a learnable `inter_scale` parameter, initialized to 1.5.

## 2.2 DISTANCE-WEIGHTED MESSAGE PASSING

Using the distance-dependent edge weights, we perform message passing at the atom and residue levels so that closer neighbors contribute more to the representation. For each target node, we compute neighbor messages with a multi-head Message Passing Neural Network (MPNN): we project source and target features into an eight-dimensional space (four heads), form explicit pairwise features via an outer product, and then aggregate these interactions with the distance-based weights. Compared to ProteinMPNN, which builds messages by just concatenating node and edge features for efficiency (Dauparas et al., 2022), we construct pairwise interactions by outer-product and then apply distance-weighted aggregation. The outer-product design is inspired by AlphaFold2's outer-product-mean operation for capturing correlations in MSA representations (Jumper et al., 2021).

## 2.3 HIERARCHICAL POOLING AND READOUT

We hierarchically pool representations from atoms to residues, and from residues to chain and complex nodes. We use the distance-based weights (defined as the sum of incoming edge weights) during pooling so that spatially proximal interactions are emphasized at higher levels. We then concatenate the pooled residue-, chain-, and complex-level representations and predict an affinity score with a Multi-Layer Perceptron (MLP) regressor. We train the model with a sum of (i) Huber loss for robust regression under noisy affinity labels (Huber, 1964) and (ii) a pairwise ranking loss (cross-entropy over correctly ordered pairs) to encourage the correct ordering of $\Delta G$ (Burges et al., 2005).

# 3 EXPERIMENTS

## 3.1 DATASETS

We use PPB-Affinity (Liu et al., 2024), which aggregates binding measurements from five sources and standardizes labels by normalizing $K_D$ (dissociation constant) and curating chain annotations, yielding 12,062 unique complexes from 3,032 PDB entries. Because the same interaction can appear multiple times across sources, we merge records with identical (PDB, chain pair, mutation) identifiers and average their affinity values. To reduce label inconsistency further, we relabel mutant entries: for a fixed complex (PDB, chain pair), reported wild-type affinities can vary across mutation records. We therefore compute the mean wild-type free energy $\overline{\Delta G}_{\mathrm{wt}} = \frac{1}{N_{\mathrm{mut}}} \sum_{i=1}^{N_{\mathrm{mut}}} \Delta G_{\mathrm{wt}}^{(i)}$ and use it as a reference to relabel each mutant as $\Delta G_{\mathrm{mut}}^{\mathrm{new}} = \overline{\Delta G}_{\mathrm{wt}} + \Delta \Delta G_{\mathrm{mut}}$.

For both wild-type and mutant complexes, we generate input structures with Boltz-2 (Passaro et al., 2025). We discard cases that fail structure prediction (e.g., memory limits) or cannot be processed by ATOMICA, resulting in 9,815 complexes (4,414 wild types and 5,401 mutants). Following the PPB-Affinity (Liu et al., 2024), we split complexes into five folds by PDB code to avoid structural leakage across folds. Finally, we apply ligand-receptor swapping as a symmetry augmentation, since exchanging the two chains does not change the binding affinity.

## 3.2 MODEL PERFORMANCE

We evaluate absolute binding affinity prediction using Spearman's Rank Correlation Coefficient (SRCC), Pearson Correlation Coefficient (PCC), and Root Mean Squared Error (RMSE) (Table 1). We compare against two unsupervised baselines (Boltz-2 inter-chain predicted TMscore (ipTM) and ProteinMPNN score) and a supervised baseline (PPB-Affinity). For ProteinMPNN, we use the sequence log-likelihood conditioned on the wild-type structure. For unsupervised scores, we apply a linear transformation to match the scale of $\Delta G$. Overall, unsupervised baselines correlate weakly with measured affinities, whereas supervised training substantially improves all metrics. HierBind performs best, achieving SRCC 0.659, PCC 0.674, and RMSE 2.161.

Table 1: Comparison to existing baselines on absolute binding affinity prediction.

| Category | Pretraining | Model | SRCC | PCC | RMSE |
|---|---|---|---|---|---|
| Unsupervised | Structure prediction | Boltz-2 ipTM | 0.184 | 0.114 | 3.874 |
| Unsupervised | Inverse folding | ProteinMPNN score | 0.163 | 0.173 | 3.790 |
| Supervised | N/A | PPB-Affinity | 0.646 | 0.656 | 2.233 |
| Supervised | Denoising, masking | HierBind | **0.659** | **0.674** | **2.161** |

Table 2: Ablation study of model components.

| | Edge | | Node | | | | |
|---|---|---|---|---|---|---|---|
| Setting | Message passing | Dist. weighting | Atom | Residue | Chains | Complex | SRCC |
| No pretraining | ✓ | ✓ | ✓ | ✓ | ✓ | ✓ | 0.619 |
| Regressor only | ✗ | ✗ | ✗ | ✗ | ✗ | ✓ | 0.180 |
| No message passing | ✗ | ✗ | ✓ | ✓ | ✓ | ✓ | 0.493 |
| No dist. weighting | ✓ | ✗ | ✓ | ✓ | ✓ | ✓ | 0.646 |
| Residue only | ✓ | ✓ | ✗ | ✓ | ✗ | ✗ | 0.603 |
| No atom | ✓ | ✓ | ✗ | ✓ | ✓ | ✓ | 0.604 |
| No chains | ✓ | ✓ | ✓ | ✓ | ✗ | ✓ | 0.640 |
| No complex | ✓ | ✓ | ✓ | ✓ | ✓ | ✗ | 0.645 |
| HierBind (full) | ✓ | ✓ | ✓ | ✓ | ✓ | ✓ | 0.659 |

### 3.3 ABLATIONS

Table 2 summarizes the contribution of each component. Using only pretrained complex embeddings with a regressor head performs poorly (SRCC 0.180), suggesting that global pretrained features alone are insufficient. The removal of message passing – equivalent to using pretrained embeddings and node type embeddings – also yields suboptimal performance (SRCC 0.493), highlighting the necessity of explicit graph reasoning. Distance weighting further improves performance. For node-level signals, residue-only modeling is competitive (SRCC 0.603), but adding higher-resolution atomic features provides the largest gain (SRCC 0.604 vs. 0.659).

### 3.4 THE EFFECTS OF RELABELING

As shown in Table 3, relabeling consistently improves performance, with SRCC increasing from 0.648 to 0.659. This supports that relabeling removes measurement noise and provides more robust targets, allowing the model to better capture physically meaningful affinity values.

Table 3: Model performance by relabeling.

| Dataset | SRCC | PCC | RMSE |
|---|---|---|---|
| Original | 0.648 | 0.664 | 2.214 |
| Relabeled | 0.659 | 0.674 | 2.161 |

## 4 CONCLUSION

This work proposes HierBind for absolute binding affinity prediction directly from protein complex structures. The key contribution is a multi-scale representation that explicitly connects atomic and residue-level interactions to higher-level summaries, enabling the model to integrate local interactions with global multimer geometry. We further improve physical fidelity with distance-weighted message passing and pretrained features, yielding an energy estimate for ranking complexes. Em-

pirically, HierBind outperforms a strong baseline on PPB-Affinity, achieving SRCC 0.659, PCC 0.674, RMSE 2.161. Ablations confirm that message passing, hierarchical modeling, and pretrained features each improve the performance. Since we use predicted structures from Boltz-2, future work could explore robustness to structural noise and prediction errors. We expect this hierarchical, multi-scale formulation to enable more reliable affinity ranking for downstream protein design.

## MEANINGFULNESS STATEMENT

We consider a meaningful representation of life to be a compact, compositional description that (i) reflects the physicochemical constraints governing biomolecular structure and interactions and (ii) supports reliable generalization to unseen sequences and structures. In this view, meaning emerges when representations capture physically grounded, multi-scale determinants of function rather than dataset-specific correlates. Our work contributes by learning a hierarchical, distance-aware representation of protein complexes that explicitly links atomic and residue-level interactions to chain-, and complex-level summaries. This yields a more faithful estimator for binding affinity.

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
