# OpenReview forum: "Hierarchical Multi-Scale Modeling of Absolute Binding Affinity in Protein Complexes"
_ICLR.cc/2026/Workshop/LMRL — ICLR 2026 Workshop LMRL Poster_

### Official Review · Reviewer_vhmq · 2026-02-11
**Review for HierBind**

**Rating:** 8
**Confidence:** 2

**Review:**

## Summary
The authors built a Hierarchical multi-scale Message Passing Neural Network (HierBind) and relabeling to model protein binding affinity. It is evaluated on a compiled benchmark, while an informative ablation study provided.

## Strengths
1. A multi-scale network can model protein from atom- to chain- level.
2. Informative ablation study provided and compared to different models, where the model achieves improvement.

## Weaknesses
1. The structure is generated by Boltz-2 for benchmark. Discussing how pLDDT affects performance may be a valuable discussion.
2. If the authors plan to extend this abstract to a regular paper, I recommend to compare with more supervised methods, also add generalization experiments.

## Clarity
1. Add how relabeling is operated in appendix may improve clarity.

---

### Official Review · Reviewer_f9fQ · 2026-02-20
**The authors propose a hierarchical approach for predicting absolute affinity by modeling complexes at 4 scales: atom, residue, chain and complex. They relabel the targets to have consistent \delta G which shows improvements in performance. Comparison to the baseline models shows the proposed approach achieves state of the art on affinity prediction.**

**Rating:** 5
**Confidence:** 4

**Review:**

The motivation behind creating hierarchical graph comes from complexes that can be integrated at different levels. Thea authors do a good job explaining the figures, dataset creation and ablations. The message passing architecture and distance weighted message passing is justified. Below are some comments

1. Is the model sensitive to the choice of k-nearest neighbors, 8 bins and the inter_scale parameter of 1.5?
2. Would the performance change if the model size was increased/decreased?
3. Can the authors compare to other structure-aware geometry based models such as GearBind/AtomSurf?
4. While one dataset is fine, it would be nice to see performance on another affinity dataset such as Sabdab
5. How much is the model sensitive to the errors from predicted structures by Boltz-2? It would be a nice comparison to show performance on experimental structures or structures from two predicted models such as AF3 and Boltz2.
6. Can the model distinguish between strong and weak binders?

If the authors could address some of the comments it would make the claim of the paper stronger.

---

### Meta-Review · Area_Chair_LdiM · 2026-02-27

**Recommendation:** Accept (Poster)
**Confidence:** 4

**Metareview:**

Accept.

---

### Decision · Program_Chairs · 2026-03-02

**Decision:**

Accept (Poster)

**Comment:**

Please see the meta-review.